# Unpaired Image-to-Image Translation with Density Changing Regularization

**Shaoan Xie**[1], **Qirong Ho**[2], and **Kun Zhang**[1,2]

[1] Carnegie Mellon University
[2] Mohamed bin Zayed University of Artificial Intelligence
shaoan@cmu.edu, qirong.ho@mbzuai.ac.ae, kunz1@cmu.edu

## Abstract

Unpaired image-to-image translation aims to translate an input image to another domain such that the output image looks like an image from another domain while important semantic information are preserved. Inferring the optimal mapping with unpaired data is impossible without making any assumptions. In this paper, we make a density changing assumption where image patches of high probability density should be mapped to patches of high probability density in another domain. Then we propose an efficient way to enforce this assumption: we train the flows as density estimators and penalize the variance of density changes. Despite its simplicity, our method achieves the best performance on benchmark datasets and needs only $56 - 86\%$ of training time of the existing state-of-the-art method. The training and evaluation code are avaliable at `https://github.com/Mid-Push/Decent`.

## 1 Introduction

Unpaired image-to-image translation aims to translate an input image to another domain such that the output image looks like an image from another domain while important semantic information are preserved. For example, in the task selfie→anime, we need to translate the selfie photo into anime style while the identity of input human is still preserved. In addition, many research problems can be reformulated as image translation tasks, such as domain adaptation [18, 34], medical image analysis [2] and image super-resolution [50].

In unpaired image translation setting, we are given two collections of samples without pairing information and we need to learn a proper mapping from one domain to another. Unfortunately, given two marginal distributions, there can be infinite number of joint distributions that can derive the same marginals [30]. Therefore, assumptions are needed to address this ill-posed problems. One popular assumption is the cycle consistency [55, 26, 48], which assumes that the proper mapping is one-to-one. Cycle consistency has been shown to achieve impressive visual performance in many tasks [55, 25]. But sometimes the one-to-one assumption can be over restrictive, especially for some tasks that one domain has more information than another one [39]. As an alternative, contrastive learning based image translation is gaining wide attention [39]. CUT [39] employs infoNCE loss [43] to maximize the mutual information between two corresponding patches in the input and output images. Recent image translation methods are mostly trying to improve CUT, such as negative sample mining [44, 24], informative sample mining [20, 51] and dual embedding learning [16].

In this paper, we propose a different way to find the optimal mapping by looking at the neighboring information. DistanceGAN [5] proposes to maintain the pairwise distance between images after the mapping. But the pairwise distance is usually computed within mini-batches and the neighboring information can be inaccurate, which may leads to unsatisfactory performance as reported in [39]. We

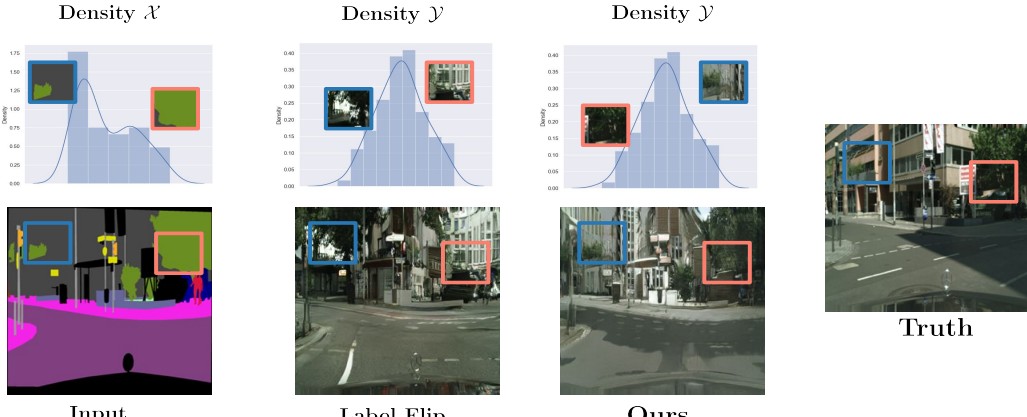

Figure 1: Illustration of our constraint. The plotted densities are synthesized for illustration. On the top row, we show the patch density distribution of domain $\mathcal{X}$ and $\mathcal{Y}$. We assume that the patches of high density should still be mapped to patches with high density in another domain.For example, patches of building (blue box) in the input image are of higher density than patches of trees (orange box) in domain $\mathcal{X}$. Then we assume that the corresponding patch should also have higher density. If label flipping happens, buildings (high density) are mapped to trees (low density) and trees (low density) are mapped to buildings (high density), the variance of density changes can be very large. So we penalize the variance of density changes to find the optimal mapping. The ground truth photo is in line with our assumption.

propose to consider the neighboring information from the density perspective: the probability density of an image or patch (a smaller piece of the image) is a good representation of its neighborhood. If the probability density of an image is high, it means that many samples are likely to be drawn from its neighborhood. In light of this analysis, we propose our density changing regularization: *patches with high (low) density should be mapped to patches with high (low) density in another domain.* As illustrated in Figure 1, the gray area in input image (blue box) is of higher density than the green area (orange box), then our regularization encourages that the gray area is mapped to area with high density in another domain. In this example, the patch of building is of higher density than the patch of trees. Therefore, our regularization encourages the gray area to be mapped to the building and the green area to be mapped to the trees. As we can observe from the ground truth, it is exactly what the ground truth photo does. The regularization is implemented in a very simple way: we just train two density estimators and penalize the variance of the density changes of patches. We apply our method to different image translation tasks and the superior performance across tasks demonstrate the effectiveness of our method. In addition, our method needs only around 56-86% training time of state-of-the-art (SOTA) methods. From the successful experimental results, we suggest that our method can be safely applied when preserving the neighboring information is needed, e.g., label→city task. However, the method may be less effective when preserving the neighboring information is not necessary. For example, for the horse→zebra task, the densities of horse patches are lower than the densities of zebra patches. So enforcing the regularization may be over strong on this specific task. In the future, we may consider applying attention on important patches only to address this issue.

## 2    Related Work

**Image Translation** For unpaired image-to-image translation, we are given only two collections of images and we need to learn a proper mapping from one domain to another. However, the number of possible mappings can be infinite and we need to introcue additional assumptions to reduce it. Cycle consistency [55, 48, 26] is one of the most popular assumptions. It enforces the mapping function to be one-to-one and has shown success in many tasks [25, 1, 8, 7, 3, 41, 46]. However, it may be restrictive when the optimal mapping is not one-to-one, e.g., segmentation map to a real photo. To address this issue, CUT [39] proposes a contrastive learning based method. It maximizes the mutual information between two corresponding patches in the input and output image with the infoNCE

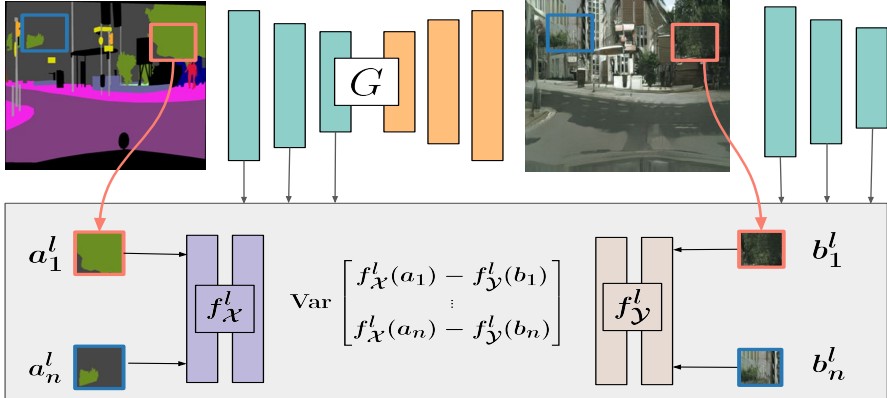

Patch Density Changing Regularization

Figure 2: Patch Density Changing Regularization for unpaired image translation. $G$ is the generator, $a_l^n$ and $b_l^n$ denotes the patch representation extracted by the $l$-th layer of $G$ of the input image $x$ and generated image $G(x)$, respectively. $f_{\mathcal{X}}^l$ and $f_{\mathcal{Y}}^l$ are log-density estimators for patch representations at $l$-th layer in domain $\mathcal{X}$ and $\mathcal{Y}$, respectively. We penalize the variance of log-density changes such that patches with high (low) density are mapped to patches with high (low) density in another domain.

loss [37]. Unlike CycleGAN [55], CUT doesn't need two mapping function and allows one-sided training. However, as many recent method revealed, the effectiveness of the method can be hampered by the negative samples used. Randomly selected negative samples might not be useful. Therefore, recent methods are mostly focusing on mining more meaningful negative samples or positive samples for better results [24, 44, 51, 20]. Unlike the mutual information based methods, our method focus on the neighboring information of patches and no negative sample are needed. There are also some methods arguably based on the neighborhood information. DistanceGAN [5] proposes to maintain the pairwise distance between images in the source domain and the translated domain. HarmonicGAN [52] proposes a smoothed distance objective inspired by [56]. F-LSeSIM [54] proposes to encourage the pairwise distance matrix of input and translated patch representations to be consistent. However, the pairwise distance loss may be inaccurate due to the insufficient sample and the training objective may also be dominated by the very distant pairs. In contrast, our method utilizes the probability density, which represents the neighborhood information of the whole dataset and we also avoid the possibly bad influence by the very distant pairs. There are also some powerful alternatives, such as the spatial transformation consistency [12, 47] and shared latent space assumption [30, 22, 29].

**Flow+GAN** FLOW-GAN [15] uses the normalizing flow as the generator. Then the generator can be trained by maximum likelihood and adversarial objectives hybridly. Based on FLOW-GAN, Alignflow[14] first translates the source image into a latent variable with flow and generate its corresponding image in the target domain by feeding the latent variable into another flow. Since flows are designed to be invertible, the mapping function is exactly invertible while CycleGAN [55] only enforces that property approximately. iFlowGAN [9] proposes to regularize the lipschitz norm of each layer and thus the mapping function is invertible and the inverse can be obtained by fixed-point theorem. Unfortunately, the flow model can take very long time to train and has inferior performance compared to GAN model in high-resolution image modeling. [32, 4, 27]. Our model doesn't need to train the flow as image modeling but we utilize its strong density estimation ability to regularize the GAN training.

**Density Estimation** Estimating the density $P(x)$ from a set of samples $\{x_i\}$ is an important problem in machine learning and computer vision. There have been many successful works that estimate the density with neural networks [38, 42, 28, 11, 21, 23, 31, 13, 45, 36, 19]. In this paper, we consider the autoregressive normalizing flow as our density estimator. The flow model $f$ maps the data into Gaussian noise and the density of data $x$ can be computed as $p(x) = p(z)|\det J_{f(x)}|$, where $z = f(x)$, $p(z)$ is the normal distribution density and $|\det J_{f(x)}|$ is the absolute value of determinant of the Jacobian matrix. With careful design of the architecture of the function $f$, the determinant can also be computed efficiently.

## 3 Proposed Method

Given images $\{x_i\} \in \mathcal{X}$ and images $\{y_j\} \in \mathcal{Y}$, our goal is learn a mapping $G : \mathcal{X} \rightarrow \mathcal{Y}$ such that $G(x)$ looks like images in the domain $\mathcal{Y}$ while preserving necessary semantic information. For example, in translating a segmentation map to real-world photo, we need to ensure that generated photo looks realistic while the contents (e.g., building, trees, roads) in the generated photo follow the segmentation map.

### 3.1 Baseline GAN model

To achieve the first goal that $G(x)$ looks like an image in another domain $\mathcal{Y}$, we employ an adversarial loss with the domain discriminator $D$ to match the distribution between $P_{G(\mathcal{X})}$ and $P_{\mathcal{Y}}$,

$$\mathcal{L}_{\text{gan}} = \mathbb{E}_{x \sim \mathcal{X}}[\log(1 - D(G(x)))] + \mathbb{E}_{y \sim \mathcal{Y}}[\log(D(y))]. \tag{1}$$

We also additionally introduce an identity mapping loss to regularize the mapping function $G$, which is widely adopted in existing image translation methods [55, 39].

$$\mathcal{L}_{\text{identity}} = \mathbb{E}_{y \sim \mathcal{Y}}\|G(y) - y\|. \tag{2}$$

By combing these two losses, we obtain an baseline GAN model for image translation. Unfortunately, there can still be infinite number of mappings that satisfy these objectives and we need further constraints to reduce the number of possible mappings in the space [55].

### 3.2 Proposed Density Constraint

In this paper, we consider the neighboring information from the probability density perspective. We first revisit the definition of density function: Let $X$ be a continuous real-valued random variable. A *density function* for $X$ is a real-valued function $f$ which satisfies

$$P(a \leq X \leq b) = \int_a^b f(x)dx, \tag{3}$$

for all $a, b \in \mathbb{R}$.

Given a value $x$ of $X$, we have $P(x - \delta \leq X \leq x + \delta) = \int_{x-\delta}^{x+\delta} f(x)dx \approx 2\delta f(x)$. For a predefined $\delta$, if the density of $x$ is high, then $P(x - \delta \leq X \leq x + \delta)$ is high. It means that there are will be many samples that fall within the range $[x - \delta, x + \delta]$. Therefore, there will be many neighbors of $x$ in the sample space. It follows that the density $f(x)$ can be viewed as a measure of how many neighbors (within $\delta$ distance) $x$ will have.

In light of this observation, we propose our density changing constraint for unpaired image-to-image translation: *if the density $f(x)$ of a image $x$ is high in the domain $\mathcal{X}$, then the translation $G(x)$ should also have high density $P(G(x))$ in domain $\mathcal{Y}$.* The density function provides us an elegant way to utilize the neighboring information without computing the time-consuming and possibly inaccurate pairwise distances. Given two density estimators $f_{\mathcal{X}}, f_{\mathcal{Y}}$ for two domains, we have our constraint defined as

$$\mathcal{L}_{\text{density}} = \mathbb{V}\left(\frac{f_{\mathcal{X}}(X)}{f_{\mathcal{Y}}(G(X))}\right), \tag{4}$$

where $\mathbb{V}(.)$ is the variance function. Variance is a smooth function that depicts the variance of a random quantity as a function of its mean. If density changes of some patches are too high or too low, the variance can be large. It means that we would like the density changes for all images to be close such that image with high density is still mapped to high density while image with low density still mapped to low density. In the following sections, we provide an efficient way to get the density estimators $f_{\mathcal{X}}, f_{\mathcal{Y}}$ and the density changing objective $\mathcal{L}_{\text{density}}$.

### 3.3 Density Changing Regularized Unpaired Image Translation

To compute $\mathcal{L}_{\text{density}}$, we need density estimators $f_{\mathcal{X}}$ and $f_{\mathcal{Y}}$. There are many successful autoregressive flows for density estimation [21, 10]. However, density estimation for images is still an challenging

task due to the high-dimensionality of images. Inspired by the recent patch-level image translation method [39], we propose to compute the densities in the patch level rather than image level. We first extract the feature maps at layer $l$ of the generator $G$ as $m^l(x) = G^l(x) \in \mathbb{R}^{c \times h \times w}$, where $c$ is the number of channels of the feature map and $h, w$ represents the height and width of the maps. Then we reshape $m^l(x)$ into patch representations $P^l(x) \in \mathbb{R}^{hw \times c}$. For each representation of size $c$, it corresponds to a patch in the input image. And we have $hw$ patch representations. We denote the patch representations of input image $x \in \mathcal{X}$ as $P^l(x) = \{a_i^l\}$, input image $y \in \mathcal{Y}$ as $P^l(y) = \{c_i^l\}$ where $i$ is the index of patch representations, generated image $G(x)$ as $P^l(G(x)) = \{b_i^l\}$. We sometimes use $a^l$ to denote the random variable whose values are $a_i^l$ and similar for $b^l$ and $c^l$.

For each layer $l$, we employ two density estimators $f_{\mathcal{X}}^l$ and $f_{\mathcal{Y}}^l$, which are parameterized as auto-regressive flows [10]. Given patch representations of real images $\{a_i^l\}$ and $\{c_i^l\}$, we train the density estimators with maximum likelihood estimation as

$$\mathcal{L}_{\text{nll}}(f_{\mathcal{X}}) = \sum_{l=1}^{L} \sum_{i=1}^{hw} -f_{\mathcal{X}}^l(a_i^l), \quad \mathcal{L}_{\text{nll}}(f_{\mathcal{Y}}) = \sum_{l=1}^{L} \sum_{i=1}^{hw} -f_{\mathcal{Y}}^l(c_i^l), \quad \mathcal{L}_{\text{nll}} = \mathcal{L}_{\text{nll}}(f_{\mathcal{X}}) + \mathcal{L}_{\text{nll}}(f_{\mathcal{Y}}), \quad (5)$$

where the outputs of the flows $f_{\mathcal{X}}$ and $f_{\mathcal{Y}}$ are log-probabilities and we need to minimize above loss to maximize the log-likelihood.

Given the density estimators, we can compute our density regularization easily as

$$\mathcal{L}_{\text{patch-density}} = \sum_{l=1}^{L} \mathbb{V}(f_{\mathcal{X}}^l(a^l) - f_{\mathcal{Y}}^l(b^l)). \quad (6)$$

The outputs of density estimators are log-likelihoods and therefore, $f_{\mathcal{X}}^l(a^l) - f_{\mathcal{Y}}^l(b^l)$ is equivalent to the probability ratios in the log-scale since $\log \frac{p}{q} = \log p - \log q$. Please note that $a^l$ and $b^l$ represents the corresponding patches in the input image $x$ and output image $G(x)$. By minimizing $\mathcal{L}_{\text{patch-density}}$, we are encouraging the transformation from $a^l$ to $b^l$ should satisfy our assumption, i.e., patches with high density should be mapped to patches with high density.

**Full Objective** Our full objective for the generator consists of three terms

$$\mathcal{L}_{\text{full}}(G) = \mathcal{L}_{\text{gan}} + \lambda_{\text{idt}} \mathcal{L}_{\text{identity}} + \lambda_{\text{density}} \mathcal{L}_{\text{patch-density}}, \quad (7)$$

where $\lambda_{\text{idt}}$ and $\lambda_{\text{density}}$ are hyper-parameters that balance different losses.

For the discriminator, we train it to maximize the $\mathcal{L}_{\text{gan}}$ to distinguish the real from the generated images:

$$\mathcal{L}_{\text{dis}}(D) = -\mathcal{L}_{\text{gan}}. \quad (8)$$

For the density estimators, we train them with $\mathcal{L}_{\text{nll}}$ as

$$\mathcal{L}_{\text{nll}}(\{f_{\mathcal{X}}^l, f_{\mathcal{Y}}^l\}_{l=1}^{L}) = \mathcal{L}_{\text{nll}}. \quad (9)$$

For brevity, we denote our method as DEnsity Changing rEgularized uNpaired image Translation (DECENT).

## 4 Implementation

**Density Estimator**. In our initial experiments, we consider three auto-regressive neural flows for density estimator: MAF [38], BNAF [10] and NSF [11]. But we observe that MAF and NSF sometimes reports NAN error, which may be caused by the exponential scaling function. BNAF further adopts the weight normalization technique to stablize the training. Therefore, we stick to BNAF in our experiments. More details are provided in the supplementary.

**Generator and Discriminator**. We follow the networks used in CUT [39]. For the generator, we adopt the 9-block ResNet as the backbone network. For the discriminator, we adopt the 3-layer PatchGAN discriminator. The learning rate is 0.0002 with $\beta_1 = 0.5$, $\beta_2 = 0.999$. The exception is the cat→dog task, which requires large shape changes. Following NEGCUT[44], we replace the patchGAN discriminator with a 6-layer discriminator, which captures more global information.

**Hyper-parameters**. We mostly follow the hyper-parameters in CUT [39]. We set $L = 5$ which is number of feature layers we use following CUT [39]. It means that we use five layers (0,4,8,12,16th) of the generator to extract patch representations. The layers correspond to receptive fields of sizes $1\times1$, $9\times9$, $15\times15$, $35\times35$, and $99\times99$. We also use 256 patches in each layer instead of all patches to save computation time and memory. We set $\lambda_{idt} = 10$ and $\lambda_{density} = 0.01$ across all tasks.

**Training** In each iteration, we first update the density estimators by minimizing $\mathcal{L}_{nll}$. Then we update the discriminator by $\mathcal{L}_{dis}$. When optimizing the generator, to compute $\mathcal{L}_{patch\text{-}density}$, we use the polyak averaged version of BNAF to estimate the density for patches. For $\mathcal{L}_{gan}$, we replace the vanilla GAN loss with the LSGAN [33] objective following CUT [39]. For all dataset, we resize images to $256\times256$. We keep the learning rate for the first half of the training and linearly decay it to 0 in the last half of training.

## 5 Experiments

### 5.1 Evaluation

We follow the evaluation protocols in [55, 39, 44, 20, 51] and run all tasks once. We report results of more runs in the supplementary material.

**Datasets** We follow the evaluation protocols in CUT [39] by running experiments on three benchmark datasets: label→city, cat→dog and horse→zebra. We further run experiment on selfie→anime dataset to fully verify the effectiveness of our method. We run label→city and horse→zebra for 400 epochs, cat→dog and selfie→anime for 200 epochs.

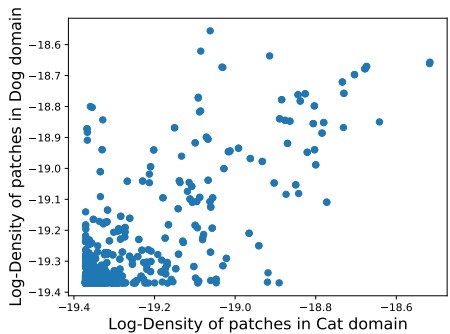

Figure 3: Visualization of the high correlation between densities in cat→dog dataset.

**Metrics** For the label→city task, we follow [39] and evaluate the generated photos by a pretrained segmentation model DRN [49]. The DRN model translates the generated photos into segmentation labels and then we can compare the output labels with the input labels. We compute mean average precision (mAP), pixel-wise accuracy (pAcc) and average class accuracy (cAcc). We find that different pretrained DRN model can cause huge difference in the output results, so we evaluate all methods and report the results by our code for fair comparison.

To evaluate the visual quality of generated images, we also adopt the widely-used Frechet Inception Distance (FID) [17] and Kernel Inception Distance (KID) [6] scores. FID and KID both measure the distribution divergence between the generated images and the real images.

**Baselines** For the first three tasks, we compare our method with different assumptions: cycle consistency-CycleGAN [55], pairwise distance preservation – DistanceGAN and SelfDistance [5] and mutual information maximization – CUT[39], DCLGAN [16], FSeSIM [54], NEGCUT [44], MoNCE [51] and QS-Attn [20]. For selfie→anime, we compare with CycleGAN [55], U-GAT-IT [25], CUT[39], CouncilGAN [35], ACL-GAN [53] and SpatchGAN [40]. For all tasks, we also run the Base-GAN model, i.e., $\lambda_{density} = 0$, to fully examine the effectiveness of our proposed regularization.

### 5.2 Justification of the Assumption

Our assumption states that patches of high density should be mapped to patches of high density in another domain. Therefore, it is important to justify whether such assumption holds on various datasets. For label→city dataset, we have ground truth pairs. For other three datasets, we use the most recent method generations as pseudo pair data, i.e., we use generation by QS-Attn on cat→dog and horse→zebra dataset and generation by SpatchGAN on selfie→anime dataset. Then we randomly crop the pair of images to patches on the same location and apply PCA to reduce the dimension. Then

| Method | label→city | | | | cat → dog | H → Z | Speed |
| | mAP ↑ | pAcc ↑ | cAcc ↑ | FID ↓ | FID ↓ | FID ↓ | Sec/it ↓ |
|---|---|---|---|---|---|---|---|
| CycleGAN [55] | 22.37 | 57.27 | 29.86 | 66.7 | 146.85 | 76.9 | 0.171 |
| DistanceGAN [5] | 7.86 | 42.74 | 11.72 | 79.0 | 143.64 | 72.1 | **0.035** |
| SelfDistance [5] | 8.25 | 43.61 | 12.20 | 58.4 | 108.3 | 78.9 | 0.036 |
| AttentionGAN [41] | - | - | - | - | - | 68.6 | - |
| CUT [39] | 27.79 | 70.70 | 35.90 | 56.4 | 76.2 | 45.5 | 0.137 |
| DCLGAN [16] | 27.75 | 68.19 | 36.72 | 49.4 | 61.6 | 42.2 | 0.225 |
| FSeSim [54] | 24.74 | 61.47 | 33.02 | 58.1 | 87.8 | 43.4 | 0.060 |
| NEGCUT [44] | 28.63 | 72.29 | 36.03 | 48.5 | 55.9 | **39.7** | 0.275 |
| MoNCE [51] | 29.12 | 72.35 | 36.48 | 54.7 | - | 41.9 | 0.231 |
| QS-Attn [20] | 29.75 | 71.76 | 37.95 | 50.2 | 80.6 | 42.0 | 0.182 |
| Base-GAN | 21.86 | 53.85 | 28.81 | **47.4** | 140.1 | 40.1 | 0.082 |
| DECENT (Ours) | **30.97** | **72.93** | **39.33** | **47.4** | **55.2** | 41.3 | 0.154 |
| Testing Images | 45.98 | 77.57 | 55.99 | 0 | 0 | 0 | - |

Table 1: Main experiment results. Abbreviations: (H)orse and (Z)ebra.

| Method | selfie→anime | |
| | FID ↓ | KID ↓ |
|---|---|---|
| CycleGAN [55] | 92.4 | 2.99 |
| DistanceGAN [5] | 94.5 | 2.36 |
| U-GAT-IT [25] | 94.8 | 2.71 |
| CUT [39] | 87.2 | 2.55 |
| CouncilGAN [35] | 92.4 | 2.65 |
| ACL-GAN [53] | 98.0 | 2.85 |
| SpatchGAN [40] | 83.3 | 2.14 |
| Base-GAN | 89.5 | 3.09 |
| DECENT (Ours) | **80.7** | **1.42** |
| Training Images | 76.7 | 0.30 |

| Setting | | label→city | | |
| Model | $\lambda$ | mAP ↑ | pAcc ↑ | cAcc ↑ |
|---|---|---|---|---|
| Base-GAN | 0 | 21.86 | 53.85 | 28.81 |
| DECENT | 0.001 | 28.15 | 64.82 | 35.40 |
| | 0.01 | **30.97** | **72.93** | **39.33** |
| | 0.1 | 30.29 | 71.51 | 38.67 |
| | 0.5 | 23.77 | 66.80 | 31.42 |
| PatchDist | 0.001 | 25.12 | 59.93 | 34.12 |
| | 0.01 | 22.10 | 54.75 | 29.73 |
| | 0.1 | 28.43 | 64.67 | 38.47 |
| | 0.5 | 25.87 | 67.87 | 33.92 |

Table 3: Ablation Study

we fit a kernel density estimator for each domain. Finally, we can obtain pair of densities for paired image patches and we compute Pearson correlation coefficient between the two sets of densities. The results are shown in Table. 4. We can observe that p-values for all datasets are 0, suggesting that the null hypothesis that the densities are uncorrelated can be rejected safely. The correlation coefficients are also significantly greater than 0 and we can arrive at 0.837 on cat→dog dataset. We also visualize the estimated log-densities of cat→dog dataset in Figure. 3. for each dot in the figure, its coordinate is determined by log-densities of a paired image patches in two domains. We can observe a clear positive correlation between the densities. This encouraging result well supports our assumption.

## 5.3 Comparison against Baselines

We present the quantitative results in Table. 1 and 2. Among the four tasks, our method achieves the best performance on the label→city, cat→dog tasks and selfie→anime tasks. In particular, our method is the first one to improve mAP above 30 on the label→city task. Please note that all recent state-of-the-art methods are improved versions of CUT [39]. By contrast, we propose an effectiveness regularization which is orthogonal to the contrastive learning [39] and cycle consistency losses [55]. Our method

| Dataset | Coefficient | p-value |
|---|---|---|
| label→city | 0.540 | 0.0 |
| cat→dog | 0.837 | 0.0 |
| horse→zebra | 0.511 | 0.0 |
| selfie→anime | 0.779 | 0.0 |

Table 4: Pearson linear correlation between densities of (pseudo) paired image patches.

brings substantial improvements over the Base-GAN model. For the label→city task, we outperform it by 41.7% on mAP, 35.4% on pAcc and 36.5% on cAcc. The significant improvements demonstrates that our proposed density regularization is able to preserve important semantic information in inputs. On the cat→dog task, our method improves the FID from 140.1 to 55.2. On the selfie→anime task,

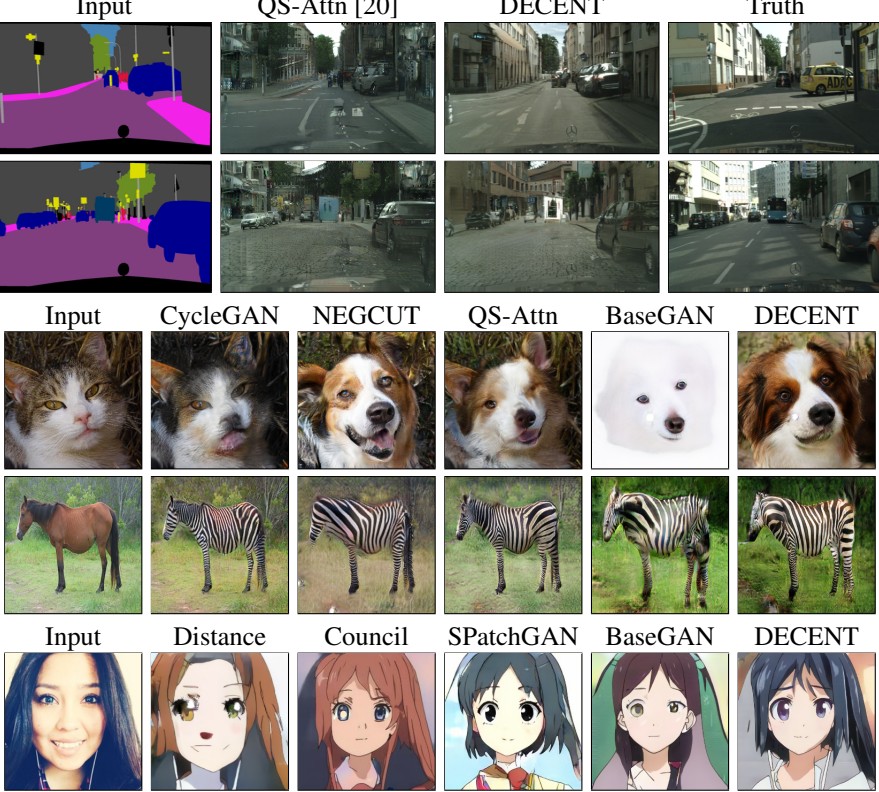

Figure 4: Visualization of samples generated by different methods: label→city, cat→dog, horse→zebra and selfie→anime. We provide more samples and more baseline results in the Supplementary material.

our method improves the FID from 89.5 to 80.7 and KID from 3.09 to 1.42. The encouraging results suggest that our regularization is also helpful in distribution matching.

We also present the generated samples in Figure. 4 and more samples are provided in the supplementary material. For the label→city task, we can observe that the SOTA method– QS-Attn [20] still faces some label flipping issue, i.e., mapping the gray area to the trees. By contrast, our method is able to map the gray area into the building accurately. For the cat→dog task, we can observe that the Base-GAN model suffers the well-known mode collapse issue while our method is able to avoid it by our proposed regularization. For the selfie→anime, our method learns to preserve the hair color of the first input person while keeping the headphone cable in the output image. Unlike CouncilGAN [35] and SpatchGAN [40], our method doesn't apply additional data augmentations, such as random brightness and hue offset.

### 5.4 Analysis

**Sensitivity of $\lambda_{\text{density}}$.** Although our method only introduces one loss, it is still important to investigate the sensitivity of the introduced hyper-parameter $\lambda_{\text{density}}$. We run our method on label→city task since the testing images are paired. The results are shown in Table. 3. We can observe that our method is robust to the scale of $\lambda$. When $\lambda = 0.01$ or $0.1$, we can achieve state-of-the-art results. When $\lambda = 0.0001$, it still improves the Base-GAN model from 21.86 to 28.15 on mAP metric. When we increase it to 0.5, which is quite far from the optimal value, we can still observe large improvements over the Base-GAN model.

**Pairwise relationship vs Density** We can observe that the performance of DistanceGAN and SelfDistance [5] are unsatisfactory in Table. 1. As we have argued in section 2, it may be caused by insufficient samples. Therefore, we build a stronger version of DistanceGAN – Patchdist. Patchdist is trained to preserve the pairwise distance between all patch representations now. Since many patches

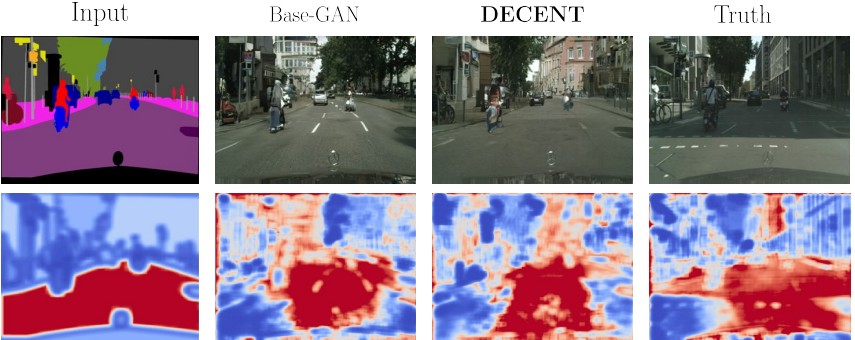

| Input | Base-GAN | **DECENT** | Truth |

Figure 5: An example of the learned densities (best viewed in color) on label→city task. Densities increases as the color transition from blue to red. We provide the learned densities for more samples and tasks in the supplementary.

are available in each iteration, the pairwise relations should be more accurate. As shown in Table. 3, PatchDist can also bring improvements over Base-GAN model now (DistanceGAN and SelfDistance are worse). The improvements supports our argument that the neighboring information is very useful in image translation tasks. Our method still outperforms PatchDist by a large margin, which implies that the densities values are more informative than the pairwise relation in image translation tasks.

**Training Time** We run all methods on NVIDIA Tesla V100-SXM2 GPU and report their training speed in table 2. One may notice that DistanceGAN [5], SelfDistance [5] and FSeSim [54] are faster than the Base-GAN model. The main reason is that they don't apply the identity regularization. When compared to the three SOTA methods (NEGCUT [44], MoNCE [51] and QS-Attn [20]), our method achieves the best performance with the highest training speed. NEGCUT needs to generate negative samples with additional networks, MoNCE needs to compute the entropy of all patch representations and MoNCE needs to address an optimal transport problem every iteration. In contrast, we only introduce the density estimators and avoids the heavy global information computation. As a consequence, our method only needs approximately 56%, 67% and 86% training time of NEGCUT [44], MoNCE [51] and QS-Attn [20], respectively.

**Learned densities** We visualize the learned densities in Figure.5. The densities increase as the color goes from blue to red. Firstly, we observe that our learned densities are consistent with human judgements and reveal the neighboring information accurately. For example, human can easily tell that gray patches should have more neighbors than green patches in the first row. The densities of gray patches shown in the second row are higher than the green patches.

Then we examine whether our method effectively enforces our assumption and whether it brings improvements. On the label→city task, we observe that the mapping of Base-GAN violates our assumption, i.e., Base-GAN maps the green patches (low density) to the building (high density) while gray patches (high density) to the trees (low density). As a consequence, it suffers the label-flipping issue when compared with the ground truth photo. In contrast, our method is able to find the correct mapping through our density regularization.

## 6 Limitation and Discussion

Although our method achieved state-of-the-art performance across many tasks, our method still suffers from some limitations: (1) the violation of the assumption; One may notice that the Base-GAN model achieves the second best FID on the horse→zebra task as shown in Table. 1. Although our method is slightly better than other SOTA methods, it still brings some performance degradation (40.1→41.3). This may be caused by the unmatched dataset statistics since it is reported that horse takes 18% pixel of the image and zebra takes 37% pixels in the dataset [39]. Therefore, patches of high density in the horse domain may not still be of high density. We also provide visualization and analysis in the supplementary. A possible solution can be adopting the attention module to only enforce our density regularization on the important patches. (2) patch correspondence. Using representation at patch level provides more samples than image level. But our method also faces the same limitation as patch-based methods [39, 20, 51]: the patches may not be well corresponded in some tasks. For

example, when the first domain is the front face while the second domain is the face profiles, we need to learn the rotation mapping between two domains, which may cause patch-based methods to fail. A possible solution would be considering larger patches, i.e., increase the number of layer used to extract patch representations. Large patches ought to contain more global information and may address this issue.

## 7 Conclusion

In this paper, we propose a simple yet effective method to address the ill-posed unpaired image-to-image translation problem. Departing from existing cycle and contrastive learning based methods, we propose to preserve the neighboring information of image patches from the density persepective. If the probability density of a patch is high, it is highly likely that it has many neighbors (similar patches). Then we propose our density changing regularization where patches with high density should be mapped to patches with high density in another domain. We also propose a simple implementation to achieve this assumptions. The superior performance on various benchmark datasets demonstrate the effectiveness of our method.

## Societal Impact

Image-to-image translation is a double-edged sword: On the one hand, it allows creative applications, such as the selfie→anime task and label→city. It also has great potential in related tasks, such as image super-resolution, medical image analysis and domain adaptation. On the other hand, it becomes easier to manipulate image data. In particular, DeepFakes have been used to create fake celebrity videos and fake news. How to avoid such misuse remains an important research problem.

## Acknowledgements

Kun Zhang was partially supported by the National Institutes of Health (NIH) under Contract R01HL159805, by the NSF-Convergence Accelerator Track-D award #2134901, by a grant from Apple Inc., and by a grant from KDDI Research Inc.

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
