# OpenReview forum: "Unsupervised Image-to-Image Translation with Density Changing Regularization"
_NeurIPS.cc/2022/Conference — NeurIPS 2022 Accept_

### Official Review · Reviewer_wXXZ · 2022-07-05

**Rating:** 6
**Confidence:** 4
**Soundness:** 3 good
**Presentation:** 3 good
**Contribution:** 3 good

**Summary:**

The paper proposes a solution for the unsupervised image-to-image translation problem which is based on matching the distribution of generated patches in the output domain Y to that of the input patch distribution in the source domain X. To this end, the paper proposes a loss that aims to minimize the variance between the density of patches in X and the corresponding density in domain Y. The loss is based on density estimation using an auto-regressive flow model applied on patch representation at different layers. The paper demonstrates superiority on a number of datasets in comparison to baseline methods.

**Questions:**

1. An assessment of the effect of patch size on the output.
2. An assessment of the limitations, such as examples or analysis of the cases where the assumption does not hold.

**Limitations:**

The authors adequately addressed the limitations and potential negative societal impact. However, it may be useful to further illustrate, qualitatively or visually the cases where the assumption made does not hold.

**Strengths And Weaknesses:**

Strengths:
1. The paper is well written and clearly illustrates the motivation behind the proposed solution for the unsupervised image-to-image problem.
2. The main technical contribution is the density loss. The motivation for using flow models to match the distribution of patches from the input distribution and the generated ones makes sense. I find the simple realization of the main idea through a single loss term to be a strength of the paper, especially as it leads to clear improvements.
3. The evaluation is relatively thorough, comparing to recent methods and direct alternatives on a range of datasets, both quantitatively and qualitatively.

Weaknesses:
1. The paper makes an assumption regarding the fact that patch distribution should be matched. While this is discussed in the limitations, I think it should be stated up front and discussed already in the introduction. The method will only work where the assumption holds.
2. An ablation study into the size of patches would be useful to understand their effect. In the limit, one could consider pixels or the entire image.

---

> ### Author Response · Authors · 2022-08-02
> **Response to Reviewer wXXZ**
>
> 1.  **The limitation should be stated up front and discussed already in the introduction.**
>
> Thanks for your kind suggestion. Accordingly, we also put this limitation discussion in the introduction section in line 48-53 of the updated manuscript.
>
> 2.  **An ablation study into the size of patches**
>
> This sounds like a great idea! We have run experiments on different patch representation layers.
>  And the results are as follows (we also included it as Table.1  and more details are discussed in section 3 in appendix):
>
> | PatchSize | mAP | PixAcc | ClsAcc |   |
> |-----------|-----|--------|--------|---|
> |       Base    |   21.86  |        53.85  |   28.81     |   |
> |         1  |    26.03|   62.40     |    34.24     |   |
> |          9  | 28.33    |   72.41     |    35.91    |   |
> |          15  | 24.18  |  58.80     |   32.36      |   |
> |          35  | 23.52   |  55.42     |   32.19    |   |
> |          99  | 28.07  |  67.63     |    36.30    |   |
> |          Full  | 30.97   | 72.93     |   39.3    |   |
>
>  We can observe that our regularization works on different patch sizes and when the patch size is 9, it works pretry good. If we adopt all patch layers, we can obtain the best result.
>
> 3. **An assessment of the limitations, such as examples or analysis of the cases where the assumption does not hold.**
>
> Thanks for your suggestion. Due to space constraints, we put the discussion and failure samples in Section 9 and Figure 5 in the appendix. We added this clarification in line 301 of the revised main paper.

---

### Official Review · Reviewer_zLZj · 2022-07-10

**Rating:** 3
**Confidence:** 5
**Soundness:** 1 poor
**Presentation:** 3 good
**Contribution:** 2 fair

**Summary:**

In this paper, the authors make a density-changing assumption where image patches of high probability density should be mapped to patches of high probability density in another domain. Then the authors propose an efficient way to enforce this assumption: they train the flows as density estimators and penalize the variance of density changes.

**Questions:**

See Weaknesses.

**Limitations:**

The author does not provide potential negative societal impact of their work.

**Strengths And Weaknesses:**

Strengths:
This paper is well written and easy to understand.

Weaknesses:
1. The results of mAP, pAcc, and cAcc are not consistent with those in the original QS-Attn paper.
2. The results of SWD on both Cat2Dog and Horse2Zebra should be included and compared in Table 1.
3. User study results should be provided.
4. Qualitative ablation results should be provided.
5. More attention-based GAN methods such as [1,2] should be included and compared in Tables 1 and 2.
6. From Figure 3, the proposed method does not significantly improve the performance of the label2city, horse2zebra, and anime datasets.
7. Results of more methods should be provided in Figure 3, especially the label2city and anime datasets.
8. Results of SOTA methods should be provided and compared in Figure 4.

[1]Kim, Junho, Minjae Kim, Hyeonwoo Kang, and Kwanghee Lee. "U-gat-it: Unsupervised generative attentional networks with adaptive layer-instance normalization for image-to-image translation." ICLR 2020.
[2]Tang, Hao, Hong Liu, Dan Xu, Philip HS Torr, and Nicu Sebe. "Attentiongan: Unpaired image-to-image translation using attention-guided generative adversarial networks." IEEE Transactions on Neural Networks and Learning Systems (2021).

---

> ### Author Response · Authors · 2022-08-02
> **Response to Reviewer zLZj**
>
> 1. **The results of mAP, pAcc, and cAcc are not consistent with those in the original QS-Attn paper.**
>
> We feel sorry about the confusion. The main reason is that there is no public available evaluation model for the label2city dataset (See this link  https://github.com/taesungp/contrastive-unpaired-translation/issues/138 ).  And different methods  and DRN model can cause huge differences in the results (https://github.com/taesungp/contrastive-unpaired-translation/issues/104). For example, for the 2 CVPR2022 papers (MoNCE, QS-ATTN), the values for CUT are also different  (MoNCE reports thatCUT can achieve 78.22 of PixAcc while QS-ATTN adopts the reported number 68.8 in CUT). For fair comparison,, we evaluate all methods and report the results with our code. We are also planning to publish the evaluation code upon the publication of this paper. We added some clarification in the revised version to explain the inconsistency in line 196-198 of the updated manuscript.
>
> 2. **The results of SWD on both Cat2Dog and Horse2Zebra should be included and compared in Table 1.**
>
> Thanks for your suggestion. We considered SWD in our initial paper. It is worth mentioning that as you may notice, there is no widely-adopted SWD implementation. The reported SWD results are different for different papers. For example, MoNCE reports that SWD of CUT is 32.02 while QS-Attn reports 31.5.  There also has been some problems with the reported SWD result (https://github.com/sapphire497/query-selected-attention/issues/2 ).
>
> 3& 4. **User study results and qualitative ablation results should be provided.**
>
> Thanks for your suggestion. We have included qualitative results in Figure 12 in the supplementary. We can clearly observe that the generations by Base-GAN often flips the building and tree. By contrast, our method is robust under different
> values of the hyper-parameter $\lambda$.
>
> 5. **More attention-based GAN methods such as [1,2] should be included and compared in Tables 1 and 2.**
>
> Thanks for your kind suggestion. We agree that these two papers should be included since attention-based methods are also an important branch in I2I.  We included U-GAT-IT in table 2 and Attention-GAN in table 1.
>
> 6. **From Figure 3, the proposed method does not significantly improve the performance of the label2city, horse2zebra, and anime datasets.**
>
> When compared to the Base-GAN model, our method achieves clear performance gain, which highlights the effectiveness of our proposed regularization. We admit that the performance gain is not so big when compared to most recent methods, e.g., QS-Attn (CVPR2022) and NEGCUT (ICCV2021). But it is worth noting that all of them are improved versions of CUT method while our method only contains a single regularization. Our method beats these complex SOTA methods with a simple regularization on various datasets and is the fastest method among recent methods. In particular, our method only needs approximately 56% training time of NEGCUT. We hope that our method could be a potential foundation for a set of methods and believe that further larger performance gain can be obtained by improving our regularization in the future.
>
> 7.  **Results of more methods should be provided in Figure 3, especially the label2city and anime datasets.**
>
> Due to the space limitations, we are unable to put more results on the main paper. We put more samples in the appendix, please check figure 6-10.
>
> 8. **Results of SOTA methods should be provided and compared in Figure 4**
>
> Thanks for your suggestion. We have added the results by QS-Attn in Figure 1 and 11 in the supplementary.  Without our density changing regularization, the SOTA method QS-Attn still generates low-density objects (tree) in
> the high density region (building).  The reason is that there is no explicit density changing regularization as in our method.

---

> > ### Comment · Reviewer_zLZj · 2022-08-09
> > **Looking forward to your answers**
> >
> > >1. The results of mAP, pAcc, and cAcc are not consistent with those in the original QS-Attn paper.
> >
> > This is a bit tricky, please upload your evaluation code, I want to check it out.
> >
> > >2. The results of SWD on both Cat2Dog and Horse2Zebra should be included and compared in Table 1.
> >
> > It seems that the evaluation metrics of this task are not very uniform, so how to judge the performance of the proposed model?
> >
> > >3 & 4. User study results and qualitative ablation results should be provided.
> >
> > No user study results were provided. Because the evaluation metrics are not uniform, the user study may be the best evaluation method. However, the author did not provide the user study results. Moreover, Figure 12 in the supplementary material is just the ablation study results for the hyper-parameter lambda. What I would like to see more is the ablation study results of the proposed model, which means I want to know which part of the model has the most impact on performance and training speed. The authors do not provide experimental results in this regard.
> >
> > >5. More attention-based GAN methods such as [1,2] should be included and compared in Tables 1 and 2.
> >
> > Why not add the results of U-GAT-IT in Table 1 and the results of Attention-GAN in Table 2？
> >
> > >6. From Figure 3, the proposed method does not significantly improve the performance of the label2city, horse2zebra, and anime datasets.
> >
> > It is difficult to recognize that the results generated by the proposed method are significantly better than other methods, which can be highlighted in the figure.
> >
> > >7. Results of more methods should be provided in Figure 3, especially the label2city and anime datasets.
> >
> > Because the authors use their own evaluation codes to evaluate all the methods in Table 1. Therefore, the author has the visualization results of all the methods in Table 1. Then why does Figure 6 in the supplementary material only compare with QS-Attn, Figure 7 compare with only MoNCE, and Figures 8, 9, and 10 only compare with four existing methods? Why not compare with all the methods in Table 1? I guess that these results may be carefully selected, and the method proposed in this paper is not comparable to some methods in some tasks, so the author makes a selective comparison.
> >
> > >8. Results of SOTA methods should be provided and compared in Figure 4.
> >
> > What I want to see is the result of comparing all the methods in Table1.
> >
> > ``Since the author did not address my concerns well, I keep the original score.``

---

> > > ### Author Response · Authors · 2022-08-09
> > > **Response**
> > >
> > > 1.  **the evaluation code**
> > >
> > > Sure! We have uploaded our evaluation code to https://github.com/anonymous-aisubmission/Neurips2022
> > > We also provide the generated results by our method, QS-Attn and MoNCE in the zip file.  Reviewer can also verify the results by downloading the pretrained model checkpoints from QS-Attn and MoNCE.
> > >
> > >
> > > 2. **how to evaluate**
> > >
> > > So we report the FID scores as in previous papers. We checked the results of previous method with the FID and the results are consistent with the numbers they report.  We use the public torch-fidelity package.
> > >
> > > 3&4. **user study and qualitative results**
> > >
> > > Sorry for the confusion, but we don't have enough time to provide you the results now. We will add them in our updated paper.
> > >
> > > 5&8. **why U-GAT-IT and AttentionGAN not in two tabless**.
> > >
> > > The reason is that we use the numbers reported the paper. The result of U-GAT-IT is taken from SpatchGAN and the result of AttentionGAN is taken from the original paper.  U-GAT-IT only provides checkpoints for selfie2anime, so we cannot include it in table 1.
> > > AttentionGAN provides results of selfie2anime in KID (the KID protocol is same as U-GAT-IT but different from the KID in SpatchGAN).  If needed, we can download the checkpoints of AttentionGAN and report more results in Table 1 and 2.  We also would like to remind that we already included the most recent methods (CVPR2022) in table 1 and 2 while U-GAT-IT was published in ICL2020 and Attention-GAN is published in 2021.
> > >
> > > 6. **highlight the difference**
> > >
> > > Thanks for your suggestion. We will add the highlight in the figure. But for label2city, we think the difference is significant. The QS-Attn clearly frequently flips the building and tree as shown in the main paper and supplementary.
> > >
> > > 7. **why not compare all methods in the same figure**
> > >
> > > The reason is that the images in label2city are much more complex than images in other dataset. If we put all methods in the same figure, users have to zoom in to tell the difference. We put four images in each row just for the ease of reading. We provide enough examples for comparison in the supplementary.  We beat SOTA methods quantatively and qualitatively. We provide the 500 generated results by our method, QS-Attn and MoNCE in the above github link. Feel free to compare them.

---

> > > ### Author Response · Authors · 2022-08-10
> > > **More results of Attention-GAN**
> > >
> > > Hi, we tried to download the checkpoints of selfie2anime of AttentionGAN to add the results in Table 2, but unfortunately we encounter the size mismatch problem, which is the same as https://github.com/Ha0Tang/AttentionGAN/issues/20
> > >
> > > We have included the results of horse2zebra in table 1. But the pretrained models for label2city and cat2dog are unavailable right now.  So we are unable to report them right now.
> > >
> > > Thanks,
> > > Authors

---

> ### Author Response · Authors · 2022-08-08
> **Looking forward to your feedbacks**
>
> Dear Reviewer  zLZj:
>
> Thanks a lot for your efforts in reviewing this paper. We tried our best to address the mentioned concerns. Are there unclear explanations here? We could further clarify them.
>
> Best, Authors.

---

### Official Review · Reviewer_o4oL · 2022-07-11

**Rating:** 5
**Confidence:** 4
**Soundness:** 2 fair
**Presentation:** 3 good
**Contribution:** 3 good

**Summary:**

This paper proposes a density changing regularization for better image translation under the assumption that the regions of the high probability density of two domains must be mapped to each other. To do so, the authors define a density constraint as the variance of the density ratio of two domains. Here, they train density estimators for each domain, numerically compute the value, and minimize it to reduce the ratio gap.

**Questions:**

Please see the above comments.

**Limitations:**

Please see the above comments.

**Strengths And Weaknesses:**

The idea is simple and effective. When the assumption holds, the results show that the density changing regularization improves the translation performance.


My major concern with this paper is its main assumption. In fact, I could not agree with the argument that the patches with a high (low) density in one domain should be mapped to patches with high (low) density in another domain. I could think of many trivial examples that this is violated (as also noted by the authors). As the authors also noted in the last section, the key to the proposed method is whether the assumptions are met. However, despite its importance, there is no justification for this assumption and no theoretical or empirical observations are supported. The authors simply state the assumption and use it without any explanation, leaving a big logical jump. There must be some experiments or analyses to validate this assumption (at least in part, for the datasets used in the paper). Or, one must verify this with enormous amounts of experiments in various benchmark datasets so readers can agree that this constraint would generally work in many practical situations. If this is not resolved, I cannot give high scores.

Another issue is the confusing usage of the terminology; “unsupervised”. In fact, this is an “unpaired” setup, not “unsupervised”. Many recent papers have addressed this misusage, the first of which is TUNIT in ICCV 2021. Please revise the term accordingly.


The subscript index “i” is used without defining it. One needs to wait until seeing eq (4) to know what it is.

There are several typos in the paper:

Line 104, there are can -> there can

Line 147, {b_i^l}-> {c_i^l},

Line 153 Please note that a^l and c^l represents -> b^l represents.

Line 189 L_mle -> L_nll

Line 285, Table 2 -> Table 1

It is not clear if the density estimators are trained together or separately. Looking at line 189, it seems like the estimators are trained together. Please clarify this more. If this is the case, wouldn’t this be too unstable? What happens in the early phase of the density estimator working poorly?

Figure 4 and its description in the “learned densities” paragraph are not easy to understand. How should it be read? It is not clear what the densities under the input semantic maps (first column, second row) in Figure 4 mean (similarly for Truth, last column, second row).

--
After the rebuttal, I changed my score from borderline reject to borderline accept.

---

> ### Author Response · Authors · 2022-08-02
> **Response to Reviewer o4oL**
>
> 1. **Justification of the assumption is needed**
>
> Thanks for your great suggestion!  We added statistical tests to justify our assumption. For the label2city dataset, we have ground truth pairs. For other datasets, we use generation by most recent method as pseudo pairs. Then we randomly crop paired images to get paired image patches. Then we fit kernel density estimators for each domain. For each pair of image patch $(x,y)$ cropped from the same location in pair images $(X,Y)$,  we feed them into nonparametric gaussian kernel density estimators $f_x$ and $f_y$.  Then we can obtain a pair of densities $(f_x(x), f_y(y))$.  Finally, we  compute Pearson correlation coefficients for the two sets of estimated densities.
>
> We observed that p-values for all datasets are 0, which allows us to safely reject the null hypothesis that the densities of patches are uncorrelated. The coefficients are significantly greater than 0 across datasets (0.540 for label2city, 0.837 for cat2dog, 0.511 for horse2zebra, 0.779 for selfie2anime), which highlights the positive correlation between the densities of paired image patches.  We also provide visualization in the revised version. Please check section 5.2 of the updated manuscript for more details. In summary, the statistical tests well align with our assumption.
>
> We agree that there are cases that our assumption can be violated, as discussed in section 6. At the same time,  from our successful experimental results and statistical tests, we humbly believe that our method works well when preserving neighboring information is needed while this problem has not been well addressed by existing cycle consistency and contrastive learning methods. We have explicitly claimed the possibility of violating our assumption in some problems in line 48-53, in the updated manuscript.
>
> 2. **Typos and term “unsupervised”**
>
> Thank you very much for sharing it! We have changed the term and corrected the typos
>
> 3.  **density estimator training**
>
> The density estimators are trained on patch representations and the gradient will not flow back to the patch representations.  Only when we compute the density changing loss, the gradient flows back to the generator network. Therefore, training the estimators is just a likelihood maximization problem and is stable.  In addition, we use the exponential moving average (EMA) of the estimator when computing the density changing loss. The EMA model produces a more stable gradient for our regularization.
>
> 4.  **how to read figure 4**
>
> For figure 4, it shows the learned densities. For each image patch representation extracted by the CNN generator, we can obtain a density by feeding it to the density estimator. Then we can trace back which patch in the input image generates this representation. Therefore, we can compute the density for each patch in the input image. For the patch with red colors, it means the estimated densities for this patch to be high (Line 282).

---

> > ### Comment · Reviewer_o4oL · 2022-08-09
> > **I raise my score to borderline accept.**
> >
> > Regarding the violation of the main assumption, I am still not completely convinced because it seems too vulnerable. In addition, I don't think this can be applied to more general conditions with various objects, which are partly shown by the lowest correlation of "label2city" dataset. Still, the authors empirically showed that this assumption works in practice, which I also value.
> >
> > Regarding figure 4, I still do not get this. In line 284, the authors say that "For example, human can easily tell that gray patches should have more neighbors than green patches in the first row. The densities of gray patches shown in the second row are higher than the green patches."
> >
> > Does this mean that the gray patches with lighter sky blue have a higher density than the green patches with (very slightly darker) sky blue? If it is, please change the example to make this clear, giving much contrast (e.g., compare them to purple patches (road), which are colored red, showing much higher density) What is the first column image of the second row? Is that also the output of the density estimation or a ground truth? (Similarly for the last one) How should it be ideally? Please clarify this in the main content.
> >
> > All in all, although I am not fully happy with the soundness of the main assumption, the authors have addressed my concern and showed that it works in practice. Thus, I raise my score from borderline reject to borderline accept.

---

> > > ### Author Response · Authors · 2022-08-09
> > > **The assumption**
> > >
> > > Thanks! Your suggestion of the justification of the assumption really benefits our paper a lot! We are also glad that your concerns have been addressed to some extent.
> > >
> > > Regarding the coefficient,  the Pearson correlation measures the linear correlation, so the real (non-linear) correlation should be stronger between the two domains across datasets, which is also demonstrated by the successful experiments.
> > >
> > > We agree that the assumption can be violated sometimes. But it is nearly impossible to get a very general assumption that works across all datasets since we are only given two marginal distributions. For example, the preliminary paper "CycleGAN" also faces failure when the optimal mapping is not one-to-one (e.g., the label$\rightarrow$city dataset in our paper).  We believe that our method is a good complement to existing i2i methods and the density chaning regularization can be further combined with other methods to handle very complex scenes.
> > >
> > > As for the Figure 4, yes, the densities decreases from from red to the blue, so the density of light blue (building) is higher than the density of dark blue (tree).  Since we have to compute the density changes, so we also have to estimate densities in the source domain (first row, first column). Yes, the second row, first column shows the estimated density of the input image.  We can observe that the road is of the highest density while the density of building (light blue) is higher than the density of tree (dark blue).  So the estimated densities of source domain is in line with human. We will adjust the contrast and add a color bar to indicate this. Thanks!

---

> ### Author Response · Authors · 2022-08-08
> **Looking forward to your feedbacks**
>
> Dear Reviewer o4oL:
>
> Thanks a lot for your efforts in reviewing this paper. We tried our best to address the your concern of the assumption. We would highly appreciate it if you could provide some feedbacks on our added justifications.
>
> Best, Authors.

---

### Official Review · Reviewer_Wucz · 2022-07-14

**Rating:** 7
**Confidence:** 3
**Soundness:** 4 excellent
**Presentation:** 4 excellent
**Contribution:** 4 excellent

**Summary:**

The paper introduced a novel method for unsupervised image-to-image translation with density changing regularization. Different from most recent work, which either assume a cycle consistency or employ contrastive learning, the paper proposed a simple yet efficient method that adds density distribution regularization atop a simple GAN model. More specifically, the density changing regularization is to minimize the change of patch density distribution after translation, where the distribution is estimated from generator layers. The paper provides examples and numerical results which show improvements over baselines on both quality of the images and training speed.

**Questions:**

1. Why choose variance as the distance function between distributions instead of other functions such as KL divergence? It would be good to see the how KL divergence would affect the model
2. The equation on line 121 has no number.
3. The deductions from line 126 to line 133 is not super clean to me, could you provide more details in appendix? I also recommend moving this part to appendix.
4. What is L in line 180?
5. The description of PatchDict in table 3 is too brief. Could you add more details in appendix?


**Limitations:**

The authors have addressed the limitations and potential negative societal impact of their work.

I would suggest authors to investigate potential solutions in the future works.

**Strengths And Weaknesses:**

Strengths:
1. The paper is generally well written. The authors provide detailed explanation on methodology and results.
2. The proposed method is simple yet efficient. Such method can be easily replicated to build a strong baseline.
3. The proposed method shows improvement over baselines on most the tasks.

Weaknesses:
1. The results on horse-to-zebra task is not as good as SOTA model. The authors provide potential issue and solution.
2. The ending of the paper is abrupt. While section 6 includes decent discussion on potential issues and solutions, a proper conclusion section is still recommended.

---

> ### Author Response · Authors · 2022-08-02
> **Response to Reviewer Wucz**
>
> Thanks a lot for your very helpful suggestion!  We hope that followings comments and the revision in our paper have addressed your concerns.
>
> 1.  **Including the Conclusion**.
>
>  We added section 7 as a conclusion in our revised version, to address your concern.
>
> 2.  **move deduction from Line 126-133 to appendix**
>
> We agree this short paragraph may not be enough to explain it. We added more explanation and moved it to the appendix now (Line 2-11).
>
> 3. **what is** $L$
>
> It is the number of layers we use to extract the patch representations. We added the explanation in the revised one (Line 175).
>
> 4. **More explanation of PatchDist**
>
> DistanceGAN proposes to preserve pairwise distance within images. Then we build a	stronger baseline – PatchDist which is trained to preserve pairwise distance within image patches. The major difference between our method and PatchDist is that our method uses density information while PatchDist uses pairwise distance as a proxy for neighboring information.The clear performance gain suggests that the density function contains more information that is relevant to our task than pairwise distance quantity.
>
> 5. **Why choose variance rather than KL**
>
> Our assumption states that image patches of high density should also be mapped to patches with low density.  Therefore, we would like to control the density changes for each patch to be close. Variance function is a good and direct yet simple measure that depicts the variance of a random quantity as a function of its mean. If the density changes of some patches are too high or too low,  the variance can well reflect it and minimizing the variance will encourage the density changes for each patch to be close. Computing the KL divergence between the distribution of density changes and the dirac distribution whose value is the mean of density changes is another way to achieve the goal. But we have to estimate the distribution of density changes  and it may cause extra computation cost and error. We added this clarification in our revised paper (L 127).

---

### Author Response · Authors · 2022-08-02
**General Response**

Thank you all for these helpful suggestions and comments! We have updated our manuscript and supplementary accordingly.

---

### Meta-Review · Area_Chair_G8RT · 2022-08-26

**Recommendation:** Accept
**Confidence:** Less certain

**Metareview:**

This paper addresses the density-mismatch problem in image-to-image translation by introducing a patch-wise variance constraint regularization. The approach is simple and effective, according to the reviewers. There were some general concerns about the validity of the assumption, but the authors appear to have sufficiently addressed those concerns. I would encourage the authors to make it clear that this is an inductive bias that they're relying on to make their method work: it's a valuable contribution but I think it's worth being extra clear that this is a reasonable assumption they built into their model but it might not be the best one.

I therefore recommend acceptance of this paper to NeurIPS.

There was one negative review from zLZj that had some useful content, but the authors seemed to address those concerns fairly well. The reviewer was showed skepticism towards the method that wasn't entirely clear to me and wanted to look at the code themselves but never followed through. I wasn't terribly convinced by the score being so low after the discussion and the author rebuttals, and I don't see evidence that reviewer looked at other reviews or discussion, so I believe the score does not accurately represent the paper's quality and I will treat the score (the discussion was good) as an outlier.

Both wXXZ and o4oL did well as far as discussion and engagement.

**Award:**

No

---

### Decision · Program_Chairs · 2022-09-14

Accept